# On model selection consistency of M-estimators with geometrically decomposable penalties

**Jason D. Lee, Yuekai Sun**
Institute for Computational and Mathematical Engineering
Stanford University
{jdl17,yuekai}@stanford.edu

**Jonathan E. Taylor**
Department of Statisticis
Stanford University
jonathan.taylor@stanford.edu

## Abstract

Penalized M-estimators are used in diverse areas of science and engineering to fit high-dimensional models with some low-dimensional structure. Often, the penalties are *geometrically decomposable*, *i.e.* can be expressed as a sum of support functions over convex sets. We generalize the notion of irrepresentable to geometrically decomposable penalties and develop a general framework for establishing consistency and model selection consistency of M-estimators with such penalties. We then use this framework to derive results for some special cases of interest in bioinformatics and statistical learning.

## 1    Introduction

The principle of parsimony is used in many areas of science and engineering to promote "simple" models over more complex ones. In machine learning, signal processing, and high-dimensional statistics, this principle motivates the use of sparsity inducing penalties for model selection and signal recovery from incomplete/noisy measurements. In this work, we consider M-estimators of the form:

$$\underset{\theta \in \mathbf{R}^p}{\text{minimize}}\ \ell^{(n)}(\theta) + \lambda\rho(\theta),\ \text{subject to } \theta \in S, \tag{1.1}$$

where $\ell^{(n)}$ is a convex, twice continuously differentiable loss function, $\rho$ is a penalty function, and $S \subseteq \mathbf{R}^p$ is a subspace. Many commonly used penalties are *geometrically decomposable*, *i.e.* can be expressed as a sum of support functions over convex sets. We describe this notion of decomposable in Section 2 and then develop a general framework for analyzing the consistency and model selection consistency of M-estimators with geometrically decomposable penalties. When specialized to various statistical models, our framework yields some known and some new model selection consistency results.

This paper is organized as follows: First, we review existing work on consistency and model selection consistency of penalized M-estimators. Then, in Section 2, we describe the notion of geometrically decomposable and give some examples of geometrically decomposable penalties. In Section 3, we generalize the notion of irrepresentable to geometrically decomposable penalties and state our main result (Theorem 3.4). We prove our main result in the Supplementary Material and develop a converse result concerning the necessity of the irrepresentable condition in the Supplementary Material. In Section 4, we use our main result to derive consistency and model selection consistency results for the generalized lasso (total variation) and maximum likelihood estimation in exponential families.

## 1.1 Consistency of penalized M-estimators

The consistency of penalized M-estimators has been studied extensively. The three most well-studied problems are (i) the lasso [2, 26], (ii) generalized linear models (GLM) with the lasso penalty [10], and (iii) inverse covariance estimators with sparsity inducing penalties (equivalent to sparse maximum likelihood estimation for a Gaussian graphical model) [21, 20]. There are also consistency results for M-estimators with group and structured variants of the lasso penalty [1, 7].

Negahban et al. [17] proposes a unified framework for establishing consistency and convergence rates for M-estimators with penalties $\rho$ that are *decomposable* with respect to a pair of subspaces $M$, $\bar{M}$:

$$\rho(x + y) = \rho(x) + \rho(y), \text{ for all } x \in M, \ y \in \bar{M}^\perp.$$

Many commonly used penalties such as the lasso, group lasso, and nuclear norm are decomposable in this sense. Negahban et al. prove a general result that establishes the consistency of M-estimators with decomposable penalties. Using their framework, they derive consistency results for special cases like sparse and group sparse regression. The current work is in a similar vein as Negahban et al. [17], but we focus on establishing the more stringent result of model selection consistency rather than consistency. See Section 3 for a comparison of the two notions of consistency.

The model selection consistency of penalized M-estimators has also been extensively studied. The most commonly studied problems are (i) the lasso [30, 26], (ii) GLM's with the lasso penalty [4, 19, 28], (iii) covariance estimation [15, 12, 20] and (more generally) structure learning [6, 14]. There are also general results concerning M-estimators with sparsity inducing penalties [29, 16, 11, 22, 8, 18, 24]. Despite the extensive work on model selection consistency, to our knowledge, this is the first work to establish a general framework for model selection consistency for penalized M-estimators.

## 2 Geometrically decomposable penalties

Let $C \subset \mathbf{R}^p$ be a closed convex set. Then the *support function* over $C$ is

$$h_C(x) = \sup_y \{y^T x \mid y \in C\}. \tag{2.1}$$

Support functions are sublinear and should be thought of as semi-norms. If $C$ is a norm ball, *i.e.* $C = \{x \mid \|x\| \le 1\}$, then $h_C$ is the dual norm:

$$\|y\|^* = \sup_x \{x^T y \mid \|x\| \le 1\}.$$

The support function is a supremum of linear functions, hence the subdifferential consists of the linear functions that attain the supremum:

$$\partial h_C(x) = \{y \in C \mid y^T x = h_C(x)\}.$$

The support function (as a function of the convex set $C$) is also additive over Minkowski sums, *i.e.* if $C$ and $D$ are convex sets, then

$$h_{C+D}(x) = h_C(x) + h_D(x).$$

We use this property to express penalty functions as sums of support functions. *E.g.* if $\rho$ is a norm and the dual norm ball can be expressed as a (Minkowski) sum of convex sets $C_1, \ldots, C_k$, then $\rho$ can be expressed as a sum of support functions:

$$\rho(x) = h_{C_1}(x) + \cdots + h_{C_k}(x).$$

If a penalty $\rho$ can be expressed as

$$\rho(\theta) = h_A(\theta) + h_I(\theta) + h_{S^\perp}(\theta), \tag{2.2}$$

where $A$ and $I$ are closed convex sets and $S$ is a subspace, then we say $\rho$ is a *geometrically decomposable* penalty. This form is general; if $\rho$ can be expressed as a sum of support functions, *i.e.*

$$\rho(\theta) = h_{C_1}(\theta) + \cdots + h_{C_k}(\theta),$$

then we can set $A$, $I$, and $S^\perp$ to be sums of the sets $C_1, \ldots, C_k$ to express $\rho$ in geometrically decomposable form (2.2). In many cases of interest, $A + I$ is a norm ball and $h_{A+I} = h_A + h_I$ is the dual norm. In our analysis, we assume

1. $A$ and $I$ are bounded.
2. $I$ contains a relative neighborhood of the origin, *i.e.* $0 \in \mathrm{relint}(I)$.

We do not require $A + I$ to contain a neighborhood of the origin. This generality allows for unpenalized variables.

The notation $A$ and $I$ should be as read as "active" and "inactive": $\mathrm{span}(A)$ should contain the true parameter vector and $\mathrm{span}(I)$ should contain deviations from the truth that we want to penalize. *E.g.* if we know the sparsity pattern of the unknown parameter vector, then $A$ should span the subspace of all vectors with the correct sparsity pattern.

The third term enforces a subspace constraint $\theta \in S$ because the support function of a subspace is the (convex) indicator function of the orthogonal complement:

$$h_{S^\perp}(x) = \mathbf{1}_S(x) = \begin{cases} 0 & x \in S \\ \infty & \text{otherwise.} \end{cases}$$

Such subspace constraints arise in many problems, either naturally (*e.g.* the constrained lasso [9]) or after reformulation (*e.g.* group lasso with overlapping groups). We give three examples of penalized M-estimators with geometrically decomposable penalties, *i.e.*

$$\underset{\theta \in \mathbf{R}^p}{\text{minimize}} \ \ell^{(n)}(\theta) + \lambda \rho(\theta), \tag{2.3}$$

where $\rho$ is a geometrically decomposable penalty. We also compare our notion of geometrically decomposable to two other notions of decomposable penalties by Negahban et al. [17] and Van De Geer [25] in the Supplementary Material.

## 2.1 The lasso and group lasso penalties

Two geometrically decomposable penalties are the *lasso* and *group lasso* penalties. Let $\mathcal{A}$ and $\mathcal{I}$ be complementary subsets of $\{1, \ldots, p\}$. We can decompose the lasso penalty component-wise to obtain

$$\|\theta\|_1 = h_{B_{\infty,\mathcal{A}}}(\theta) + h_{B_{\infty,\mathcal{I}}}(\theta),$$

where $h_{B_{\infty,\mathcal{A}}}$ and $h_{B_{\infty,\mathcal{I}}}$ are support functions of the sets

$$B_{\infty,\mathcal{A}} = \left\{ \theta \in \mathbf{R}^p \mid \|\theta\|_\infty \le 1 \text{ and } \theta_\mathcal{I} = 0 \right\}$$
$$B_{\infty,\mathcal{I}} = \left\{ \theta \in \mathbf{R}^p \mid \|\theta\|_\infty \le 1 \text{ and } \theta_\mathcal{A} = 0 \right\}.$$

If the groups do not overlap, then we can also decompose the group lasso penalty group-wise ($\mathcal{A}$ and $\mathcal{I}$ are now sets of groups) to obtain

$$\sum_{g \in \mathcal{G}} \|\theta_g\|_2 = h_{B_{(2,\infty),\mathcal{A}}}(\theta) + h_{B_{(2,\infty),\mathcal{I}}}(\theta).$$

$h_{B_{(2,\infty),\mathcal{A}}}$ and $h_{B_{(2,\infty),\mathcal{I}}}$ are support functions of the sets

$$B_{(2,\infty),\mathcal{A}} = \left\{ \theta \in \mathbf{R}^p \mid \max_{g \in \mathcal{G}} \|\theta_g\|_2 \le 1 \text{ and } \theta_g = 0, \ g \in \mathcal{A} \right\}$$
$$B_{(2,\infty),\mathcal{I}} = \left\{ \theta \in \mathbf{R}^p \mid \max_{g \in \mathcal{G}} \|\theta_g\|_2 \le 1 \text{ and } \theta_g = 0, \ g \in \mathcal{I} \right\}.$$

If the groups overlap, then we can duplicate the parameters in overlapping groups and enforce equality constraints.

## 2.2 The generalized lasso penalty

Another geometrically decomposable penalty is the *generalized lasso* penalty [23]. Let $D \in \mathbf{R}^{m \times p}$ be a matrix and $\mathcal{A}$ and $\mathcal{I}$ be complementary subsets of $\{1, \ldots, m\}$. We can express the generalized lasso penalty in decomposable form:

$$\|D\theta\|_1 = h_{D^T B_{\infty,\mathcal{A}}}(\theta) + h_{D^T B_{\infty,\mathcal{I}}}(\theta). \tag{2.4}$$

$h_{D^T B_{\infty,\mathcal{A}}}$ and $h_{D^T B_{\infty,\mathcal{I}}}$ are support functions of the sets

$$D^T B_{\infty,\mathcal{A}} = \{x \in \mathbf{R}^p \mid x = D_{\mathcal{A}}^T y, \|y\|_{\infty} \leq 1\} \tag{2.5}$$

$$D^T B_{\infty,\mathcal{I}} = \{x \in \mathbf{R}^p \mid x = D_{\mathcal{I}}^T y, \|y\|_{\infty} \leq 1\}. \tag{2.6}$$

We can also formulate any generalized lasso penalized M-estimator as a linearly constrained, lasso penalized M-estimator. After a change of variables, a generalized lasso penalized M-estimator is equivalent to

$$\underset{\theta \in \mathbf{R}^k, \gamma \in \mathbf{R}^p}{\text{minimize}} \ \ell^{(n)}(D^\dagger \theta + \gamma) + \lambda \|\theta\|_1 , \text{ subject to } \gamma \in \mathcal{N}(D),$$

where $\mathcal{N}(D)$ is the nullspace of $D$. The lasso penalty can then be decomposed component-wise to obtain

$$\|\theta\|_1 = h_{B_{\infty,\mathcal{A}}}(\theta) + h_{B_{\infty,\mathcal{I}}}(\theta).$$

We enforce the subspace constraint $\theta \in \mathcal{N}(D)$ with the support function of $\mathcal{R}(D)^\perp$. This yields the convex optimization problem

$$\underset{\theta \in \mathbf{R}^k, \gamma \in \mathbf{R}^p}{\text{minimize}} \ \ell^{(n)}(D^\dagger \theta + \gamma) + \lambda(h_{B_{\infty,\mathcal{A}}}(\theta) + h_{B_{\infty,\mathcal{I}}}(\theta) + h_{\mathcal{N}(D)^\perp}(\gamma)).$$

There are many interesting applications of the generalized lasso in signal processing and statistical learning. We refer to Section 2 in [23] for some examples.

### 2.3 "Hybrid" penalties

A large class of geometrically decomposable penalties are so-called "hybrid" penalties: infimal convolutions of penalties to promote solutions that are sums of simple components, *e.g.* $\theta = \theta_1 + \theta_2$, where $\theta_1$ and $\theta_2$ are simple. If the constituent simple penalties are geometrically decomposable, then the resulting hybrid penalty is also geometrically decomposable.

For example, let $\rho_1$ and $\rho_2$ be geometrically decomposable penalties, *i.e.* there are sets $A_1, \ I_1, \ S_1$ and $A_2, \ I_2, \ S_2$ such that

$$\rho_1(\theta) = h_{A_1}(\theta) + h_{I_1}(\theta) + h_{S_1^\perp}(\theta)$$

$$\rho_2(\theta) = h_{A_2}(\theta) + h_{I_2}(\theta) + h_{S_2^\perp}(\theta)$$

The M-estimator with penalty $\rho(\theta) = \inf_\gamma \{\rho_1(\gamma) + \rho_2(\theta - \gamma)\}$ is equivalent to the solution to the convex optimization problem

$$\underset{\theta \in \mathbf{R}^{2p}}{\text{minimize}} \ \ell^{(n)}(\theta_1 + \theta_2) + \lambda(\rho_1(\theta_1) + \rho_2(\theta_2)). \tag{2.7}$$

This is an M-estimator with a geometrically decomposable penalty:

$$\underset{\theta \in \mathbf{R}^{2p}}{\text{minimize}} \ \ell^{(n)}(\theta_1 + \theta_2) + \lambda(h_A(\theta) + h_I(\theta) + h_{S^\perp}(\theta)).$$

$h_A$, $h_I$ and $h_{S^\perp}$ are support functions of the sets

$$A = \{(\theta_1, \theta_2) \mid \theta_1 \in A_1 \subset \mathbf{R}^p, \ \theta_2 \in A_2 \subset \mathbf{R}^p\}$$

$$I = \{(\theta_1, \theta_2) \mid \theta_1 \in I_1 \subset \mathbf{R}^p, \ \theta_2 \in I_2 \subset \mathbf{R}^p\}$$

$$S = \{(\theta_1, \theta_2) \mid \theta_1 \in S_1 \subset \mathbf{R}^p, \ \theta_2 \in S_2 \subset \mathbf{R}^p\}.$$

There are many interesting applications of the hybrid penalties in signal processing and statistical learning. Two examples are the huber function, $\rho(\theta) = \inf_{\theta = \gamma_1 + \gamma_2} \|\gamma_1\|_1 + \|\gamma_2\|_2^2$, and the multitask group regularizer, $\rho(\Theta) = \inf_{\Theta = B + S} \|B\|_{1,\infty} + \|S\|_1$. See [27] for recent work on model selection consistency in hybrid penalties.

## 3 Main result

We assume the unknown parameter vector $\theta^\star$ is contained in the *model subspace*

$$M := \text{span}(I)^\perp \cap S, \tag{3.1}$$

and we seek estimates of $\theta^\star$ that are "correct". We consider two notions of correctness: (i) an estimate $\hat{\theta}$ is *consistent* (in the $\ell_2$ norm) if the estimation error in the $\ell_2$ norm decays to zero in probability as sample size grows:

$$\left\| \hat{\theta} - \theta^\star \right\|_2 \xrightarrow{p} 0 \text{ as } n \to \infty,$$

and (ii) $\hat{\theta}$ is *model selection consistent* if the estimator selects the correct model with probability tending to one as sample size grows:

$$\mathbf{Pr}(\hat{\theta} \in M) \to 1 \text{ as } n \to \infty.$$

NOTATION: We use $P_C$ to denote the *orthogonal projector* onto $\text{span}(C)$ and $\gamma_C$ to denote the *gauge function* of a convex set $C$ containing the origin:

$$\gamma_C(x) = \inf_x \{\lambda \in \mathbf{R}+ \mid x \in \lambda C\}.$$

Further, we use $\kappa(\rho)$ to denote the *compatibility constant* between a semi-norm $\rho$ and the $\ell_2$ norm over the model subspace:

$$\kappa(\rho) := \sup_x \{\rho(x) \mid \|x\|_2 \leq 1, \ x \in M\}.$$

Finally, we choose a norm $\|\cdot\|_\varepsilon$ to make $\left\|\nabla \ell^{(n)}(\theta^\star)\right\|_\varepsilon$ small. This norm is usually the dual norm to the penalty.

Before we state our main result, we state our assumptions on the problem. Our two main assumptions are stated in terms of the *Fisher information matrix*:

$$Q^{(n)} = \nabla^2 \ell^{(n)}(\theta^\star).$$

**Assumption 3.1** (Restricted strong convexity). *We assume the loss function $\ell^{(n)}$ is locally strongly convex with constant $m$ over the model subspace, i.e.*

$$\ell^{(n)}(\theta_1) - \ell^{(n)}(\theta_2) \geq \nabla \ell^{(n)}(\theta_2)^T (\theta_1 - \theta_2) + \frac{m}{2} \|\theta_1 - \theta_2\|_2^2 \tag{3.2}$$

*for some $m > 0$ and all $\theta_1, \theta_2 \in B_r(\theta^\star) \cap M$.*

We require this assumption to make the maximum likelihood estimate unique over the model subspace. Otherwise, we cannot hope for consistency. This assumption requires the loss function to be curved along certain directions in the model subspace and is very similar to Negahban et al.'s notion of restricted strong convexity [17] and Buhlmann and van de Geer's notion of compatibility [3]. Intuitively, this assumption means the "active" predictors are not overly dependent on each other.

We also require $\nabla^2 \ell^{(n)}$ to be locally Lipschitz continuous, *i.e.*

$$\|\nabla^2 \ell^{(n)}(\theta_1) - \nabla^2 \ell^{(n)}(\theta_2)\|_2 \leq L \|\theta_1 - \theta_2\|_2 .$$

for some $L > 0$ and all $\theta_1, \theta_2 \in B_r(\theta^\star) \cap M$. This condition automatically holds for all twice-continuously differentiable loss functions, hence we do not state this condition as an assumption.

To obtain model selection consistency results, we must first generalize the key notion of *irrepresentable* to geometrically decomposable penalties.

**Assumption 3.2** (Irrepresentability). *There exist $\tau \in (0, 1)$ such that*

$$\sup_z \{V(P_{M^\perp}(Q^{(n)} P_M (P_M Q^{(n)} P_M)^\dagger P_M z - z)) \mid z \in \partial h_A(B_r(\theta^\star) \cap M)\}$$

$$< 1 - \tau,$$

*where $V$ is the* infimal convolution *of $\gamma_I$ and $\mathbf{1}_{S^\perp}$*

$$V(z) = \inf_u \{\gamma_I(u) + \mathbf{1}_{S^\perp}(z - u)\}.$$

If $u_I(z)$ and $u_{S^\perp}(u)$ achieve $V(z)$ (*i.e.* $V(z) = \gamma_I(u_I(z))$), then $V(u) < 1$, means $u_I(z) \in \text{relint}(I)$. Hence the irrepresentable condition requires any $z \in M^\perp$ to be decomposable into $u_I + u_{S^\perp}$, where $u_I \in \text{relint}(I)$ and $u_{S^\perp} \in S^\perp$.

**Lemma 3.3.** *$V$ is a bounded semi-norm over $M^\perp$, i.e. $V$ is finite and sublinear over $M^\perp$.*

Let $\|\cdot\|_\varepsilon$ be an error norm, usually chosen to make $\left\|\nabla \ell^{(n)}(\theta^\star)\right\|_\varepsilon$ small. $V$ is a bounded semi-norm over $M^\perp$, hence there exists some $\bar\tau$ such that

$$V(P_{M^\perp}(Q^{(n)}P_M(P_M Q^{(n)} P_M)^\dagger P_M x - x)) \leq \bar\tau \left\|x\right\|_\varepsilon \tag{3.3}$$

$\bar\tau$ surely exists because (i) $\|\cdot\|_\varepsilon$ is a norm, so the set $\{x \in \mathbf{R}^p \mid \|x\|_\varepsilon \leq 1\}$ is compact, and (ii) $V$ is finite over $M^\perp$, so the left side of (3.3) is a continuous function of $x$. Intuitively, $\bar\tau$ quantifies how large the irrepresentable term can be compared to the error norm.

The irrepresentable condition is a standard assumption for model selection consistency and has been shown to be almost necessary for sign consistency of the lasso [30, 26]. Intuitively, the irrepresentable condition requires the active predictors to be not overly dependent on the inactive predictors. In Supplementary Material, we show our (generalized) irrepresentable condition is also necessary for model selection consistency with some geometrically decomposable penalties.

**Theorem 3.4.** *Suppose Assumption 3.1 and 3.2 are satisfied. If we select $\lambda$ such that*

$$\lambda > \frac{2\bar\tau}{\tau} \|\nabla \ell^{(n)}(\theta^\star)\|_\varepsilon$$

*and*

$$\lambda < \min \begin{cases} \frac{m^2}{L} \frac{\tau}{2\bar\tau\kappa(\|\cdot\|_\varepsilon)\left(2\kappa(h_A) + \frac{\tau}{\bar\tau}\kappa(\|\cdot\|_\varepsilon^*)\right)^2} \\ \frac{mr}{2\kappa(h_A) + \frac{\tau}{\bar\tau}\kappa(\|\cdot\|_\varepsilon^*)}, \end{cases}$$

*then the penalized M-estimator is unique, consistent (in the $\ell_2$ norm), and model selection consistent, i.e. the optimal solution to (2.3) satisfies*

*1. $\left\|\hat\theta - \theta^\star\right\|_2 \leq \frac{2}{m}\left(\kappa(h_A) + \frac{\tau}{2\bar\tau}\kappa(\|\cdot\|_\varepsilon^*)\right)\lambda$,*

*2. $\hat\theta \in M := \operatorname{span}(I)^\perp \cap S$.*

*Remark* 1. Theorem 3.4 makes a *deterministic* statement about the optimal solution to (2.3). To use this result to derive consistency and model selection consistency results for a statistical model, we must first verify Assumptions (3.1) and (3.2) are satisfied with high probability. Then, we must choose an error norm $\|\cdot\|_\varepsilon$ and select $\lambda$ such that

$$\lambda > \frac{2\bar\tau}{\tau} \|\nabla \ell^{(n)}(\theta^\star)\|_\varepsilon$$

and

$$\lambda < \min \begin{cases} \frac{m^2}{L} \frac{\tau}{2\bar\tau\kappa(\|\cdot\|_\varepsilon)\left(2\kappa(h_A) + \frac{\tau}{\bar\tau}\kappa(\|\cdot\|_\varepsilon^*)\right)^2} \\ \frac{mr}{2\kappa(h_A) + \frac{\tau}{\bar\tau}\kappa(\|\cdot\|_\varepsilon^*)} \end{cases}$$

with high probability.

In Section 4, we use this theorem to derive consistency and model selection consistency results for the generalized lasso and penalized likelihood estimation for exponential families.

## 4 Examples

We use Theorem 3.4 to establish the consistency and model selection consistency of the generalized lasso and a group lasso penalized maximum likelihood estimator in the high-dimensional setting. Our results are nonasymptotic, *i.e.* we obtain bounds in terms of sample size $n$ and problem dimension $p$ that hold with high probability.

### 4.1 The generalized lasso

Consider the linear model $y = X^T \theta^\star + \epsilon$, where $X \in \mathbf{R}^{n \times p}$ is the design matrix, and $\theta^\star \in \mathbf{R}^p$ are unknown regression parameters. We assume the columns of $X$ are normalized so $\|x_i\|_2 \leq \sqrt{n}$. $\epsilon \in \mathbf{R}^n$ is *i.i.d.*, zero mean, sub-Gaussian noise with parameter $\sigma^2$.

We seek an estimate of $\theta^\star$ with the generalized lasso:

$$\underset{\theta \in \mathbf{R}^p}{\text{minimize}} \ \frac{1}{2n}\|y - X\theta\|_2^2 + \lambda \|D\theta\|_1 \,, \qquad (4.1)$$

where $D \in \mathbf{R}^{m \times p}$. The generalized lasso penalty is geometrically decomposable:

$$\|D\theta\|_1 = h_{D^T B_{\infty,\mathcal{A}}}(\theta) + h_{D^T B_{\infty,\mathcal{I}}}(\theta).$$

$h_{D^T B_{\infty,\mathcal{A}}}$ and $h_{D^T B_{\infty,\mathcal{I}}}$ are support functions of the sets

$$D^T B_{\infty,\mathcal{A}} = \{x \in \mathbf{R}^p \mid x = D^T y, y_\mathcal{I} = 0, \|y\|_\infty \le 1\}$$
$$D^T B_{\infty,\mathcal{I}} = \{x \in \mathbf{R}^p \mid x = D^T y, y_\mathcal{A} = 0, \|y\|_\infty \le 1\}.$$

The sample fisher information matrix is $Q^{(n)} = \frac{1}{n}X^T X$. $Q^{(n)}$ does not depend on $\theta$, hence the Lipschitz constant of $Q^{(n)}$ is zero. The restricted strong convexity constant is

$$m = \lambda_{\min}(Q^{(n)}) = \inf_x \{x^T Q^{(n)} x \mid \|x\|_2 = 1\}.$$

The model subspace is the set

$$\text{span}(D^T B_{\infty,\mathcal{I}})^\perp = \mathcal{R}(D_\mathcal{I}^T)^\perp = \mathcal{N}(D_\mathcal{I}),$$

where $\mathcal{I}$ is a subset of the row indices of $D$. The compatibility constants $\kappa(\ell_1), \kappa(h_A)$ are

$$\kappa(\ell_1) = \sup_x \{\|x\|_1 \mid \|x\|_2 \le 1, \ x \in \mathcal{N}(D_\mathcal{I})\}$$
$$\kappa(h_A) = \sup_x \{h_{D^T B_{\infty,\mathcal{A}}}(x) \mid \|x\|_2 \le 1, \ x \in M\} \le \|D_\mathcal{A}\|_2 \sqrt{|\mathcal{A}|}.$$

If we select $\lambda > 2\sqrt{2}\sigma\frac{\bar{\tau}}{\tau}\sqrt{\frac{\log p}{n}}$, then there exists $c$ such that $\mathbf{Pr}\left(\lambda \ge \frac{2\bar{\tau}}{\tau}\left\|\nabla\ell^{(n)}(\theta^\star)\right\|_\infty\right) \le 1 - 2\exp\left(-c\lambda^2 n\right)$. Thus the assumptions of Theorem 3.4 are satisfied with probability at least $1 - 2\exp(-c\lambda^2 n)$, and we deduce the generalized lasso is consistent and model selection consistent.

**Corollary 4.1.** *Suppose $y = X\theta^\star + \epsilon$, where $X \in \mathbf{R}^{n \times p}$ is the design matrix, $\theta^\star$ are unknown coefficients, and $\epsilon$ is i.i.d., zero mean, sub-Gaussian noise with parameter $\sigma^2$. If we select $\lambda > 2\sqrt{2}\sigma\frac{\bar{\tau}}{\tau}\sqrt{\frac{\log p}{n}}$ then, with probability at least $1 - 2\exp\left(-c\lambda^2 n\right)$, the solution to the generalized lasso is unique, consistent, and model selection consistent, i.e. the optimal solution to (4.1) satisfies*

1. $\left\|\hat{\theta} - \theta^\star\right\|_2 \le \frac{2}{m}\left(\|D_\mathcal{A}\|_2 \sqrt{|\mathcal{A}|} + \frac{\tau}{2\bar{\tau}}\kappa(\ell_1)\right)\lambda,$

2. $\left(D\hat{\theta}\right)_i = 0,$ *for any $i$ such that $\left(D\theta^\star\right)_i = 0$.*

## 4.2 Learning exponential families with redundant representations

Suppose $X$ is a random vector, and let $\phi$ be a vector of *sufficient statistics*. The exponential family associated with these sufficient statistics is the set of distributions with the form

$$\mathbf{Pr}(x;\theta) = \exp\left(\theta^T \phi(x) - A(\theta)\right),$$

Suppose we are given samples $x^{(1)}, \ldots, x^{(n)}$ drawn *i.i.d.* from an exponential family with unknown parameters $\theta^\star \in \mathbf{R}^p$. We seek a maximum likelihood estimate (MLE) of the unknown parameters:

$$\underset{\theta \in \mathbf{R}^p}{\text{minimize}} \ \ell_{\text{ML}}^{(n)}(\theta) + \lambda \|\theta\|_{2,1} \,, \text{ subject to } \theta \in S. \qquad (4.2)$$

where $\ell_{\text{ML}}^{(n)}$ is the (negative) log-likelihood function

$$\ell_{\text{ML}}^{(n)}(\theta) = -\frac{1}{n}\sum_{i=1}^n \log \mathbf{Pr}(x^{(i)};\theta) = -\frac{1}{n}\sum_{i=1}^n \theta^T \phi(x^{(i)}) + A(\theta)$$

and $\|\theta\|_{2,1}$ is the group lasso penalty

$$\|\theta\|_{2,1} = \sum_{g \in \mathcal{G}} \|\theta_g\|_2 \, .$$

It is also straightforward to change the maximum likelihood estimator to the more computationally tractable pseudolikelihood estimator [13, 6], the neighborhood selection procedure [15], and include covariates [13]. For brevity, we only explain the details for the maximum likelihood estimator.

Many undirected graphical models can be naturally viewed as exponential families. Thus estimating the parameters of exponential families is equivalent to learning undirected graphical models, a problem of interest in many application areas such as bioinformatics.

Below, we state a corollary that results from applying Theorem 3.4 to exponential families. Please see the supplementary material for the proof and definitions of the quantities involved.

**Corollary 4.2.** *Suppose we are given samples* $x^{(1)}, \ldots, x^{(n)}$ *drawn* i.i.d. *from an* exponential family *with unknown parameters* $\theta^\star$. *If we select*

$$\lambda > \frac{2\sqrt{2L_1}\bar{\tau}}{\tau} \sqrt{\frac{(\max_{g \in \mathcal{G}} |g|) \log |\mathcal{G}|}{n}}$$

*and the sample size* $n$ *is larger than*

$$\max \left\{ \begin{array}{l} \frac{32 L_1 L_2^2 \bar{\tau}^2}{m^4 \tau^4} \left(2 + \frac{\tau}{\bar{\tau}}\right)^4 (\max_{g \in \mathcal{G}} |g|)|\mathcal{A}|^2 \log |\mathcal{G}| \\ \frac{16 L_1}{m^2 r^2} (2 + \frac{\tau}{\bar{\tau}})^2 (\max_{g \in \mathcal{G}} |g|)|\mathcal{A}| \log |\mathcal{G}|, \end{array} \right.$$

*then, with probability at least* $1 - 2\big(\max_{g \in \mathcal{G}} |g|\big) \exp(-c\lambda^2 n)$, *the penalized maximum likelihood estimator is unique, consistent, and model selection consistent, i.e. the optimal solution to* (4.2) *satisfies*

1. $\left\| \hat{\theta} - \theta^\star \right\|_2 \leq \frac{2}{m} \left(1 + \frac{\tau}{2\bar{\tau}}\right) \sqrt{|\mathcal{A}|}\lambda,$

2. $\hat{\theta}_g = 0, \ g \in \mathcal{I}$ *and* $\hat{\theta}_g \neq 0$ *if* $\left\| \theta_g^\star \right\|_2 > \frac{1}{m} \left(1 + \frac{\tau}{2\bar{\tau}}\right) \sqrt{|\mathcal{A}|}\lambda.$

## 5  Conclusion

We proposed the notion of geometrically decomposable and generalized the irrepresentable condition to geometrically decomposable penalties. This notion of decomposability builds on those by Negahban et al. [17] and Candés and Recht [5] and includes many common sparsity inducing penalties. This notion of decomposability also allows us to enforce linear constraints.

We developed a general framework for establishing the model selection consistency of M-estimators with geometrically decomposable penalties. Our main result gives deterministic conditions on the problem that guarantee consistency and model selection consistency; in this sense, it extends the work of [17] from estimation consistency to model selection consistency. We combine our main result with probabilistic analysis to establish the consistency and model selection consistency of the generalized lasso and group lasso penalized maximum likelihood estimators.

## Acknowledgements

We thank Trevor Hastie and three anonymous reviewers for their insightful comments. J. Lee was supported by a National Defense Science and Engineering Graduate Fellowship (NDSEG) and an NSF Graduate Fellowship. Y. Sun was supported by the NIH, award number 1U01GM102098-01. J.E. Taylor was supported by the NSF, grant DMS 1208857, and by the AFOSR, grant 113039.

## Footnotes

[1]Given the extensive work on consistency of penalized M-estimators, our review and referencing is necessarily incomplete.

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
