[Reviews · NeurIPS 2013]

Submitted by Assigned_Reviewer_5

The paper proposes a unified framework for the *model selection consistency* (sparsistency) of M-estimators with penalties enjoying a property called geometrically decomposability (eg, generalized lasso). Under strong convexity and irrepresentability conditions, the authors show model selection consistency results for M-estimators with such penalties. The framework allows to derive theoretical results for M-estimators with constraints, such as isotonic regression in Sec. 4.2. Experimental results are presented to corroborate the theoretical results.



### Detailed comments

1) A weakness of the paper is the lack of detailed discussion of the various definitions of ``decomposability'' for lasso-like estimators, namely: Negahban et al.'s decomposable penalties, Van de Geer's weakly decomposable penalties (arXiv:1204.4813), and the geometrically decomposable penalties defined in the paper. These definitions clearly overlap for some simple M-estimators such as the regular lasso or group-lasso. Therefore, extensions are there to encompass cases such as the generalized lasso. On the hand, they also depart from each other, as for instance Negahban et al.'s framework does not seem to allow for M-estimators with linear constraints as in isotonic regression.

2) The supplemental material is also lacking in terms of review of previous works. For instance, the technique termed the ``dual certificate technique'' in the supplement was previously used in Juditsky and Nemirovski's series of papers on sparse estimators and theoretical guarantees. This is not acknowledged in the paper.
Summary: The paper is well-written and clearly fills a gap in the literature. The paper however seriously lacks a detailed discussion of previous works on the topics. In particular, the relationship with the work of Van de Geer (arXiv:1204.4813) should be clarified.

Submitted by Assigned_Reviewer_7

The paper provides a unified framework analysis on the model selection consistency with geometrically decomposable penalties. As special cases of this framework, it also derives the consistency results of some machine learning examples.

This paper deals with very interesting topic in the sense that; while the model consistencies have been already derived 'individually' by several works, there has been a recent trend to provide a unified framework on statistical guarantee for more different types of M-estimator for the future.

Nevertheless, my major concern on this paper is its clarity; it is not clearly written, and explanation is somehow terse.

- How are the "geometrically" decomposable penalties connected to the conventional decomposability in [18] and [6]? They are briefly described only in Introduction, not in Section 2 where authors introduce a geometrically decomposability. It would be helpful if the differences (if any) are compared explicitly

- The details of the proof is not provided in the paper, but it is surprising that the only irrepresentability "without the dependency condition", is needed for the model consistency, and moreover, Theorem 3.4 does not include minimum eigenvalue on the Fisher information matrix. This condition was needed even for Lasso analyzed in [25] that could be one of the simplest form. It would be instructive if authors can derive corollary from Theorem 3.4 for the simplest Lasso case, and compare it against the result in [25].

- Is that necessarily required to provide \ell_2 error bounds? I would not be surprised at all if the \ell_2 error bound is provided under the restricted strong convexity. I am curious if the rsc condition is just needed to provide \ell_2 error bound, or it is required even for \ell_infty norm? For Lasso case in [25], RE conditioned is not required. It seems more discussion would be helpful for the main theorem.


- Instead of \ell_2 bound, it would be better if \ell_infity bound is provided for model selection. In Theorem 3.4, isn't it required that the minimum value of true parameter is greater than something?

- How can the irrepresentability Eq. 3.3 be reduced for the examples in Section 4?


Minor comments:

Check Eq.2.2; two decomposed regularizers are same.
Summary: This is a potentially very interesting paper, but I believe it suffers in clarity and sufficient explanations.

Submitted by Assigned_Reviewer_8

This paper proves sparsistency for a class of M-estimators with what is called ``geometrically decomposable penalty''.

Geometric decomposability is I believe a novel concept introduced by the authors. It is similar to decomposability in the Negabhan et al. sense but specially suited for sparsistency analysis, i.e. primal dual witness. It seems that decomposable norms are also geometrically decomposable, though the authors did not elaborate on the connections between these two concepts.

The paper is well written. The concepts are clearly explained except for a few minor points. I am not clear what is the $\Phi$-norm that suddenly appeared in line 223 and Theorem 3.4. Also, it would be nice to remind the reader that $m$ is the restricted strong convexity constant in the theorems.

The results look reasonable. I have not checked the proof in detail. I think, except for the introduction of the geometric decomposability concept, the proof follows the Wainwright 2009 Sharp Thresholds paper and is formulaic; the authors should correct me if I am wrong here. The authors also use the sparsistency results to prove consistency, the process of which I think also follows standard proof techniques.

The authors could elaborate more on the results in section 4. For instance, is there any aspect of the theorems that is especially significant or interesting? Also, when one uses equation (3.3) to derive the irrepresentability condition of the generalized lasso, how does that condition compare with the condition of vanilla lasso? Does the condition imply any constraints on the matrix D? Is there a way to intuitively interpret the irrepresentability condition for the exponential family results? The condition there seems to involve only the log-partition function $A(\theta)$.

The paper also studies sparse M-estimation with homogenous linear equality constraints. The picture there seems a bit murky however; linear equality constraints should improve the rate of convergence since it provides additional information but the theorems don't explicitly show such improvements.

The experimental section is OK. The lines in Figure 1.2 do not seem to line up that well to me. Also, it'd be nice to have some experimental results on generalized lasso.

This is a fairly good paper. The sparsistency analysis does not seem very difficult or novel; the results, though not too surprising, are solid; and the notion of geometric decomposability introduced in the paper should be useful.

Some miscellaneous notes:
* type on equation (2.2), one of the ``A'' should be ``I''
* I am not sure why the authors mentioned isotonic regression. Isotonic regression has linear inequality constraints yes but it is a nonparametric method. There are $O(n)$ parameters in isotonic regression so restricted strong convexity will not hold I think; do the authors assert that their framework applies to sparse isotonic regression as well?
Summary: A theoretical paper that generalizes lasso sparsistency analysis, though the generalization does not seem too novel or significant.
Author Feedback

Author rebuttal: We thank the reviewers for their comments and detailed suggestions.
General Comments: We are glad that the reviewers find geometric decomposability a novel concept. We agree that a weakness of our paper is the lack of a detailed comparison between geometric decomposability and various alternative notions in the literature. We will include such a discussion in the camera-ready version. We will also simplify the general irrepresentable condition for each of the examples we present to illustrate the relationship between our result and existing results for specific M-estimators.
Specific Comments:
1. We will include references to Juditsky and Nemirovski series of papers on sparse estimators and van de Geer’s paper on weakly decomposable penalties.
2. We do require a restricted eigenvalue condition for our main result (see Assumption 3.1). We state our main result in terms of the \ell_2 norm, but the result can be converted to an arbitrary norm by changing the definitions of the compatibility constants and the norm in Assumption 3.1.
3. We will clarify the $\phi$-norm in the statement of the theorem.
When linear constraints are incorporated, the model subspace is reduced so the restricted eigenvalue, the irrepresentable constant, and compatibility constants all become more favorable. It is hard to quantify how much more favorable because this depends on the relative orientations of the sets A, I, and the subspace S.